# Occurrence of Mycotoxins in Foods: Unraveling the Knowledge Gaps on Their Persistence in Food Production Systems

**DOI:** 10.3390/foods12234314

**Published:** 2023-11-29

**Authors:** Sher Ali, Lucas Gabriel Dionisio Freire, Vanessa Theodoro Rezende, Muhammad Noman, Sana Ullah, Gul Badshah, Muhammad Siddique Afridi, Fernando Gustavo Tonin, Carlos Augusto Fernandes de Oliveira

**Affiliations:** 1Department of Food Engineering, Faculty of Animal Science and Food Engineering (FZEA), University of São Paulo (USP), Pirassununga 13635-900, SP, Brazil; alisher@usp.br (S.A.); lucasgdfreire@usp.br (L.G.D.F.); sanaullah@usp.br (S.U.); 2Faculty of Veterinary and Animal Science (FMVZ), University of São Paulo (USP), Pirassununga 13635-900, SP, Brazil; vanessatrezende@usp.br; 3Plant Molecular Physiology, Department of Biology, Federal University of Lavras (UFLA), Lavras 37200-000, MG, Brazil; muhammad.noman@ufla.br; 4Department of Health and Biological Sciences, Abasyn University Peshawar (AUP), Peshawar 25000, Khyber Pakhtunkhwa, Pakistan; abdullah4us.71@gmail.com; 5Department of Chemistry, Federal University of Paraná (UFPR), Curitiba 81530-000, PR, Brazil; gulbadshahhum1188@gmail.com; 6Department of Plant Pathology, Federal University of Lavras (UFLA), Lavras 37200-900, MG, Brazil; msiddiqueafridi@gmail.com; 7Department of Biosystems Engineering, Faculty of Animal Science and Food Engineering (FZEA), University of São Paulo (USP), Pirassununga 13635-900, SP, Brazil; fgtonin@usp.br

**Keywords:** mycotoxins, occurrence, food, cereals, nuts, persistence

## Abstract

In this review, the intricate issue about the occurrence levels of mycotoxins in foods is discussed aiming to underline the main knowledge gaps on the persistence of these toxicants in the food production system. Mycotoxins have been a key challenge to the food industry, economic growth, and consumers’ health. Despite a breadth of studies over the past decades, the persistence of mycotoxins in foods remain an overlooked concern that urges exploration. Therefore, we aimed to concisely underline the matter and provide possible biochemical and metabolic details that can be relevant to the food sector and overall public health. We also stress the application of computational modeling, high-throughput omics, and high-resolution imaging approaches, which can provide insights into the structural and physicochemical characteristics and the metabolic activities which occur in a stored cereal grain’s embryo and endosperm and their relationship with storage fungi and mycotoxins on a cellular level. In addition, there is a need for extensive collaborative network and funding, which will play a key role in finding effective solutions against the persistence of mycotoxins at the genetic and molecular to metabolic levels in the food system.

## 1. Introduction

Mycotoxins are toxic substances produced as secondary fungal metabolites that can cause harmful effects on humans and animals. These compounds considerably affect the food industry and public health. The occurrence and levels of many mycotoxins have been studied in the overall food system, in which such toxins are more notable in the most nutritious and largely consumed foods like cereals and nuts [1]. More than 300 types of toxic fungal metabolites have been identified, although the most relevant and known mycotoxins include the aflatoxins (AFs) (AFB_1_, AFB_2_, AFG_1_, AFG_2_), deoxynivalenol (DON), zearalenone (ZEN), ochratoxin (OTA), and fumonisins (FBs) (FB_1_, FB_2_, FB_3_, among others), which are highly stable metabolites that can resist food processing [2] and therefore remain in the final products. Consequently, various strategic pre-processing methods are adopted to reduce mycotoxins in the food chain [3]. Due to their remarkable toxicity and associated health risks, the threshold levels for mycotoxins have been regulated worldwide [2,4,5]. Numerous diagnostic tools are used to discriminate fungi by their genetic details and volatile signatures and determine mycotoxins in foodstuffs [6,7], but the issues and gaps regarding mycotoxins’ persistence in the food chain remain unnoticed.

Consistent with the variable chemical and structural characteristics, mycotoxins can react and modify proteins, carbohydrates, lipids, and metabolites within the food matrix [5,8]. These molecular interactions or modifications can directly influence the physicochemical properties of the food, which can cause negative effects on overall quality of the product. Mycotoxins’ interaction with proteins and cellular glucans has been disclosed in some special examples [9,10,11], but not in the food itself. However, extending these details further can provide valuable insights into mycotoxins’ persistence and interactions with the further chemical constituents in foods, including the cereals and nuts [9,10,11]. Besides the occurrences, mycotoxins’ persistence, and interactions in foods are concisely underlined. Special emphasis is given to the cereals and nuts industry, by providing a possible biochemistry discussion regarding toxins’ occurrence, persistence, and interaction issues that may be relevant to the food industry and the public health. The objective of this review is to explore the intricate issue about the occurrence levels of mycotoxins in foods, to underline the main knowledge gaps on the persistence of these toxicants in the food production system.

## 2. Mycotoxin Occurrence in Foodstuffs

The occurrence and contamination of fungi are major issues in the overall food system. A plethora of fungi occur and produce several toxins in the food and feed produced worldwide [12,13,14,15]. These toxins transfer to humans directly via the ingestion of plant-based foods of any category, as well as animal-derived food products. In the latter case, whenever animals are fed with a diet potentially contaminated with toxins, the animals biochemically transform these toxins or distribute the unmetabolized forms in various edible tissues, which can be found in their derived milk, meat, liver, heart, and eggs [16,17,18,19]. The consumption of mycotoxin-contaminated foods presents significant health risks in humans, considering the carcinogenic, immune-suppressive, estrogenic, and organ-target acute effects of these toxins [15,19].

The occurrence levels of several mycotoxins in a range of foodstuffs recently reported in some countries are displayed in Table 1. Mycotoxins occur in multiple food and feed items at varied levels, in which cereal grains and nuts are prominent. The origin of the products is diverse, where the data show that a high frequency and level of these toxins can occur regardless of the geographical location [20,21,22]. The products shown are associated with cereals (wheat, corn, oat), nuts (peanuts, hazelnuts), animal-derived products (milk, meat, eggs), mixed food, and others, e.g., cocoa and soybean meal (Table 1). These products (*N* = 5146) have often been analyzed by high-performance liquid chromatography (HPLC) and mass spectrometry (MS), and a predominant number (*n* = 4351) of the products mainly relate to cereals. This corroborates data presented in previous studies, where cereals with a widespread consumption were widely examined for mycotoxins [20,23]. In search for mycotoxins, around 28 different metabolites comprising their (less or un)familiar, emerging, and conjugated forms were evaluated, in which several have properly regulated and estimated tolerable limits, with toxicity details for the specific ingredient of interest as [8,24,25,26,27,28,29].

Regarding the overall samples evaluated in studies conducted between 2017 and 2022 and detailed in Table 1, higher contamination rates were observed for trichothecene type B (TTB) toxins (58.53%), FBs (51.19%), ZEN (40.25%), and AFs (17%), as depicted in Figure 1. Importantly, trichothecene type B includes DON as a major toxin, and FBs were more notable in cereal products, while ZEN was more likely mainly in animal products (chicken and dairy) when compared with its low level in other items. Relevant to the sample positivity for the occurrence of one or more toxins, TTB appeared in 87.47% of the corn-based feed, 70.56% of the wheat as food, and 53.10% of oat grass. Mixed products associated with cereals and nuts, as well as corn, exhibited the highest percentage of contamination by FBs (77.35–81.55%). For ZEN, samples that exhibited high positivity were from dairy milk (99%), chicken meat (52.17%), and corn feed (51.48%). Concerning the contamination by AFs, all peanut meal (100%), 62.5% of milk formula, and 54.29% of mixed cereal- or nut-based foods showed positivity for AFs (Figure 1).

When it comes to TTB having DON as a key toxin, its prevalence and level in cereals and cereal-derived products appear to show consistency across different regions. This corroborates Khaneghah et al. [35], who pointed out a higher occurrence and concentration of DON in cereal products worldwide. This observation is corroborated by additional studies. For instance, research from Algeria found that 33.3% of barley, corn, rice, and wheat samples had elevated levels of DON (ranging from 15–4569 µg/kg) [36] In Egypt, approximately 70.2% of the wheat and corn products had high contamination by DON (ranging up to 853 µg/kg) [37] A study in Brazil highlights widespread DON in various food items like rice (median 13.2 µg/kg), beans (51.3 µg/kg), wheat flour (408.2 µg/kg), corn flour (56.7 µg/kg), and cornmeal (51.8 µg/kg) [14]. In line with FBs, this group of toxins show maximum probability with cereal products. For example, barley, corn, rice, and wheat from Algeria presented surprisingly higher incidence and levels of FBs (up to 48,878.00 µg/kg) in 21 samples that exceeded the maximum permitted limit (MPL, 1000 µg/kg) in these foodstuffs according to the European Union (EU) [24,36] Regarding ZEN, it occurred in almost 100% of animal products while little in cereal commodities, indicating sample size variations (Figure 1). However, further studies have shown ZEN occurrence with higher rates in cereal commodities. For example, in the cereal group, rice wheat, and corn products are likely to be more contaminated with maximum incidence of ZEN. This likelihood aligns with a higher ZEN level (8–474 µg/kg) in rice, wheat bread, and pasta, as well as in corn-based cake or meal from Brazil [38]. Higher ZEN (median levels 57.9–70.9 µg/kg) occurrence was found in wheat grains, with varying frequencies (12–84%) in Brazil, where ZEN was above the MPL of the EU in numerous samples [39]. In similar cereal products from Pakistan, 57% of the samples had ZEN positivity, with 31% of samples above the MPL (50 µg/kg) [40]. The occurrence frequency of AFs was also high in the remaining products (Table 1, Figure 1). The data can be reinforced with the results from research describing the AFs most frequently occurring in these products [20]. High positivity rates were recorded for milk products (63%) and peanut meal (100%). In general, it can be noted that the contamination levels of various toxins surveyed within five years were high in many products, mainly in China [17], and in this case, a large variability may indicate the sample size. Compared to the established limits by the European Union, for example, the mean levels in 96 and 108 samples of corn from Croatia and Serbia during a four-year survey presented 11.87 µg/kg and 11.1 µg/kg of AFs that are above the established MPL for cereals [25,33]. Contrarily, a survey in Chile using 24 to 26 samples of milk formulas and fluid milk presented a percent rate for AFM_1_ positivity of 62.5% and 23.1%, respectively. The mean levels were 0.0038 and 0.0069 µg/kg, below the EU thresholds [33,41]. These details shed light on the prevalence of multi-mycotoxin contamination with varied levels in numerous products, which can be a matter of exposure that causes health issues in humans and animals [24,25,26,27,28,29].

Human exposure to these toxins based on their biomarkers in bodily fluids, e.g., urine, blood, and serum, has also been described [41,42,43]. Exposure to these toxins can demonstrate high levels of positivity compared to non-exposed volunteers. In addition, the exposed group can present biomarkers with average levels higher than non-exposed ones [44]. In epidemiological investigations, in non-occupational exposure, sample positivity percentages vary. This is in line with a study by Šarkanj et al. [45] from Nigeria, where urine samples were evaluated for AFM_1_, FB_1_, OTA, DON, ZEN, and other less usual biomarkers, highlighting sample positivity of 72%, 71%, 78%, 19%, and 82%. Contrarily, Njumbe Ediage et al. [44] found lower frequencies of biomarkers in human urine collected in Belgium, with 13%, 7%, 3%, 10%, 10%, and 10% for DON, OTα, CIT, β-ZEL, ZEN, and OTA, respectively. These results indicate that mycotoxin exposure frequency can differ according to several factors, including occupation, origin, biomarker monitoring, and exposure to toxins, mainly by ingesting contaminated food, inhalation, or other routes of exposure [42]. These details indicate mycotoxins’ persistence and interactions in the foods, which should be concisely underlined. Such effort could be emphasized mainly in the cereals and nuts industry, with some possible biochemistry discussion regarding toxins’ occurrence, persistence, and interaction issues that can be relevant to the food industry and public health.

## 3. Cereals and Nuts: Significance and Challenges

Among cereals, wheat, corn, and rice have been staple foods for decades in human diets and continue to be widely accepted. These food products contribute to approximately 60% of the world’s energy intake [46]. Their energy and nutritional attributes are facilitated by the diverse chemical composition of these cereals, enabling their widespread consumption. In terms of molecular diversity, this food group contains many essential micro- and macronutrients. These contents differ among the cereal types, with wheat, corn, and rice have varying degree of proteins, carbohydrates and starches, different lipid classes, phytosterols (phytoestrogens), and vitamins (thiamine, riboflavin, niacin, folate, etc.), as well as minerals, among others. These chemicals with different quantities are allocated in various compartments like the bran, endosperm, and further parts of the grains. While the average nutritional contents quantitatively differ, 100 g of cereals can contain 14.6% protein, 72.47 g carbohydrates, 1.92 g lipids, and 4.966 mg vitamins [47,48].

Similarly, nuts have been another food category that is vital in terms of nutritious and consumption values. The nutritive features depend mainly on nut components including proteins, various lipid classes, vitamins, and other chemical components, which cumulatively make them remarkable substitutes for animal-derived proteins [49,50]. The chemical contents also vary depending on the type of nut. Peanuts and walnuts have prominent amounts of proteins, carbohydrates, lipids, minerals, vitamins (e.g., thiamine, riboflavin, niacin, and folate), etc. The average nutritional constituents in nuts based on the quantity in peanuts and walnuts account for 19.70 g proteins, 19.55 g carbohydrates, 52.85 g lipids, and 6.65 mg vitamins [51]. Likewise, nuts contain essential elements (Fe, Ca, Mg, etc.), and with respect to the overall composition, nuts are medicinally relevant in several health conditions [52,53].

Nevertheless, cereals and nuts are the most susceptible commodities to a variety of field and storage fungi and toxin contamination, as has been emphasized (Table 1, Figure 1). It is true that fungi, and especially their toxins, are cumbersome to remove, which continuously demotes the quality of these items and likewise can promote health issues. The intrinsic tenacity of toxins in the food tissue is an unobserved but major issue. The mysterious persistent behavior of toxins should be considered with their chemical nature and interactions with the chemical components in the food tissue.

### 3.1. Fungi and Mycotoxin Persistence in the Food System

In this context, persistence refers to the ability of mycotoxins to remain stable within the food and feed tissues, and this tenacity makes them difficult to remove by any of the numerous currently used practical approaches [54]. Notably, these methods are able to alleviate but not totally remove mycotoxins, and in certain conditions, some practices can negatively influence the quality of the product. Thus, there is a need for industrial protocols that can capably remove toxins without affecting the quality. To achieve this, understanding mycotoxin persistence on a biochemical level directly in food/feed tissue is indispensable. This could help improve food security and production practices that can be safe and effective against this issue. To comprehend persistence, examining the (bio)chemistry and metabolic processes associated with toxins in model foodstuffs (cereals, nuts) pre- and post-harvest, as well as the chemical makeup of these items, is imperative. The selection of these foods is vital, as they are rich in proteins, carbohydrates, lipids, metabolites, and numerous other constituents that make them nutritionally important, yet they are most vulnerable to fungal invasion and activities (Figure 2).

The growth of fungi can be a complex process, but basically its mechanism (Figure 2) initiates with sporulation until a mature fungus is developed. In this generic mechanism, the fungus releases chemicals, notably the enzymes and metabolites, which cause damage to the food on intracellular levels. This means, using hydrolytic enzymes, fungi degrade macromolecular proteins and carbohydrates to attain nutrients and energy for their growth by exerting variation in the metabolism, including nitrogen and carbon metabolism [55]. This condition can be referred to as infection, and whenever it occurs, the defense system of the host is activated that acts against fungi in that specific microenvironment. Utilizing this opportunity, the fungus starts to generate and use its secondary metabolites or mycotoxins as defense signals [55]. Mycotoxins interact with molecules present in the tissue and remain localized in the tissue throughout the lifespan of the host. These aspects have been established for field fungi in crops [56] but there is a lack of studies exploring a similar challenge for storage fungi, especially in cereal grains and nuts. The study of such an overlooked issue on the metabolic level can offer valuable insights into identifying mycotoxins’ persistence and their detoxification directly in foods.

### 3.2. Metabolic Insights Related to the Mycotoxin Persistence in Food

Field fungi comprising *Alternaria*, *Cladosporium*, *Helminthosporium*, and *Fusarium* (e.g., *F. oxysporum*) can greatly cause a huge loss to economically valued crops, including cereals. However, *Fusarium* (e.g., *F. oxysporum*) is more studied, which primarily causes vascular wilt or browning of conductive tissues (xylem) and demise of the crop plant. In xylem, this fungus emits small effector proteins, referred to as “secreted in xylem” (Six), that contribute to its virulence. *Fusarium*, an anamorphic fungus, develops asexual spores including microconidia, macroconidia, and chlamydospores [57]. Spore germination is triggered through the exudates released by the host plant roots and laterally emerging roots or injury. On germination, the infectious hyphae start to grow, penetrate across the root’s epidermis tip, and intracellularly spread in the root’s cortical cells until reaching the xylem tissues. In these tissues, the fungus likely undergoes prolific branching and produces micro- and macroconidia that acropetally pass within transpiration pull in plants. Microconidia germinate and spread hyphae throughout the host plant. Ear infection primarily occurs during anthesis and is favored by wet weather, high humidity, and warm temperature. The dispersal mode involves mainly the production of asexual conidia spores, disseminated by the wind and rain to the wheat heads and that predominantly infect wheat ears in a short period of high susceptibility during anthesis [58]. *Fusarium* (*F. graminearum*) in its growth secrets effector proteins on infected wheat and in culture medium. These proteins act like “putative secretion signals” in infection, which thereby promote the production of mycotoxins, such as trichothecene—e.g., DON [55]. This is an opportunistic process, where the host plant is mimicked to generate the chemical defense signals, which are utilized as cues by the fungus for the synthesis of DON. The chemical signals differ according to plant variety, but involve polyamines, sugar (sucrose, 1-kestose, etc.), certain phenolic compounds (e.g., ferulic and coumaric acid), magnesium ions, moderate temperature (~25 °C), and pH, respectively [55]. Consistently, fungi contaminate the plant tissue with the production of many mycotoxins, such as toxic metabolites, which can possibly be present in the tissue according to the biochemical mechanisms in the host.

Plants are natural detoxifiers of xenobiotics, involving mycotoxins. This action takes place by both the compartmentation and chemical transformation of toxins. In the latter case, such as a phase-I reaction, a toxic substance, depending on its nature, is either oxidized, reduced, hydrolyzed, etc. and becomes open to further metabolic processes in the host system. The chemically modified molecule is further shaped into simpler or complex conjugates via the conjugation referred to as a phase-II reaction. These reactions are assisted by enzymes, counting on esterases, amidases, glucosyl-, malonyl-, glutathione-*S*-transferases (GSTs), etc. The phase-I process is more liable to the makeover of lipophilic molecules. Conjugates in plants are formed due to the reaction of glucose, malonic acid, and glutathione (GSH, γ-glutamyl-cysteinyl-glycine) residues with that of a functional group in target molecules. For example, glucosyl can couple with the hydroxy (OH), thiol (SH), amine (NH), and carboxy (COOH) groups while malonyl with the “OH” and “NH”. GSH has great affinity to the electrophilic sites in target molecules. The conjugated product is transferred from the cytosol to the vacuole or tonoplast in the plant using carriers, likely the membrane-bound transporters [56,59,60]. The carriers differ, for instance, glycosylated conjugates use a carrier coupled to a transmembrane proton (H^+^) gradient (P type ATPase) for transport in the tonoplast, while glutathionylated products use the ABC transporter(s) aided by adenosine-5′-triphosphate (ATP) [61]. GSH is more significant in plants [62] since in the conjugation mechanism it acts like a nucleophile, and in the presence of substrate-specific catalysts (GSTs), this nucleophile reacts with the electrophilic sites of the mycotoxins. In plant GSTs, serine residue is involved as an active center. GSH contains “NH”, two “COOH”, two peptide bonds, and an “SH” group, and due to these sites, the molecule turns highly polar and hydrophilic. The established conjugates of GSH with several mycotoxins can be irreversible, unable to pass through the biological membranes and freely move between compartments. The conjugations of GSH can occur with those mycotoxins that preserve either an epoxide (e.g., DON), lactone (e.g., AFs, patulin), or even molecules with an aldehyde group in their structures. Conversely to GSH, conjugates of glucose with toxins are not that stable, yet they can be inverted by the glycosidases in plants and even in animals [56]. Overall, fungi infect plant tissue with the production of many mycotoxins, which can be present in their conjugated forms in the raw product throughout the lifespan from field to processing, storage, reprocessing, and final consumption by the users.

Studies have primarily considered fungal dynamics in crop plants in the field and not necessarily in the products in storage. It is notable that recent studies only outline the contamination quantitatively and qualitatively but not the exact interactive mechanisms of toxins in tissue on the metabolic level in the stored items. In fact, the fungus can principally contaminate the product throughout its storage. In these foodstuffs, there are both the field (*Fusarium*, etc.) and storage (*Aspergillus*, *Penicillium*, etc.) fungi. The grain/nut seeds undergo nearly similar mechanisms regarding infectivity and mycotoxin production. In fact, the seed still has a living embryo that can redirect the required metabolic activities, gaseous exchange, and respiration, among other interactive events [63]. According to Wu et al. [63], storage fungus in the grains can be dormant (due to unfavorable conditions), latent (weakly interacting with external environment, with less favorable conditions), or in a resonant state (self-stimulated under fully favorable conditions). According to the study, in the latter situations, the fungus continues to develop and perform metabolic processes, along with the release of heat and water to the local environment, and these two factors cumulatively facilitate fungus growth in the material [63]. Consistent with these and several other factors, in storage, a further possibility for mycotoxin generation is that of the co-evolving competitive fungi, bacteria, insects, and, in general, the microflora in the grain. In such complex interactive networks, the co-evolved agents may be opportunistic with one another and they can be opposed under a competitive mechanism which also promotes fungal growth and mycotoxin production [64,65].

*Fusarium*, as a field fungus that produces trichothecene type B that includes DON as a key precursor, was prevalent in 58.53% of the overall samples, often in cereal products. TTB was prevalent in 87.47% of corn, 70.56% of wheat, and 53.10% of oat (Figure 1). High DON incidence may be associated with its high hydrophilicity. DON in its free state was more uniformly accumulated when compared to its less hydrophilic acetylated or glycosylated conjugates [66,67]. DON in the grains also reflects both the probability of the fungus hyphae reaching the inner part of the endosperm and toxin migration to the intact grain in the field. This suggests two scenarios: firstly, fungal hyphae can infiltrate the inner parts of the seed and, secondly, DON may diffuse within the intact grains. For DON, it might shift from the fungus surface to the host tissue before the complete invasion of the fungus, thus compromising the host initial defense against the pathogen [55]. A variable allocation of DON also indicates differences in its mobility within the plant, likely because DON and DON-3-glucoside molecules are more polar than others (such as ZEN and ZEN derivatives). This could be further supported by analyzing ergosterol, a lipophilic compound found in fungal cell membranes, and it is likely distributed in milling fractions [66]. The occurrence and levels of a free or conjugated glycosylated and acetylated DON were higher in the bran than in the endosperm of wheat grain [66]. A two-fold higher contamination of DON and DON-3-glucoside was described in the grains, where the levels of these toxins were likely higher in the bran. In contrast to the free DON, 3- or 15-acetylated DON shows maximum affinity to the starchy endosperm of the grain [66]. The intracellular chemical components within the food can be transduced as signals by the food content, in which lipids are more considerable. This is because cereals and nuts in their tissues preserve several types of lipids, which provide a base to mimic various metabolic processes including the transduction and signaling pathways [68]. Cellular lipids, for example, oxylipin, proactively perform in the accumulation of toxins in the food tissue’s components, as shown for DON in wheat, barley, and others [69]. It has also been described that DON can exert oxidative stress damage that leads to the generation of reactive oxygen species (ROS) in these environments [69].

The incidence shown was common for several mycotoxins in almost all items (Table 1). AFs, being the products of storage *Aspergillus* species (e.g., *A. flavus* and *A. parasiticus*), were usual in every type of food or feed (Figure 1). The incidence of AFs was visible in 100% of peanut meal, 62.5% of milk formula, and 54.29% of mixed cereals and nut products. From the chemical perspective, these metabolites are polar and counter both extreme pH (<3 or >10) and oxidizing agents. AFs can react with substrates with somewhat similar properties and with oxidants in a certain medium. Depending on the conditions, AFs may perform both reversible and irreversible reactions using their lactone moiety. Consistent with the structural trends, AFB_1_ has unsaturation at the terminal ring in one of the fused furans that can act as a chemically reactive functional group. This functional site allows this metabolite to react with ROS in the medium, which ultimately generates an unstable but reactive intermediate product—AFB_1_-epoxide. This intermediate epoxide causes additional reactions to form, for example, “AFB_1_–DNA adduct” and other chemical species [70] that lead to undesirable results. This is an enzymatically driven oxidation such as the phase-I mechanism in the grain in which a reactive “AFB_1_-epoxide” is formed that acts like an “electrophile” and reacts with the nucleophilic sites (e.g., nitrogen “N”, oxygen “O”, sulfur “S”, etc.) of another cellular molecule in a target cereal grain. This is in line with a study [70] showing that, for example, the AFB_1_ can conjugate to the nucleophilic “N” of the cellular “DNA molecule” in the grain, which leads to mutations or damage in the cell. This is reinforced by another study, where AFB_1_ was associated with key abnormalities or alterations in the meiotic cellular chromosome and 5SrDNA in durum wheat seedlings [71]. Moreover, cereals (e.g., oat, barley, etc.) are a major source of energy due to carbohydrates, and these involve “β-glucan” polymers. Glucans are polysaccharides that consist of “D-glucose” monomers polymerized via β-1,3 and β-1,4 glycosidic linkages. They vary in their structures, functions, and reaction mechanisms within the food [72]. We believe that, for example, AFB_1_ reacts and makes stable conjugated products with these moieties in the food matrix. This is to be confirmed since almost similar glucan moieties from specific yeasts are used as a decontaminant strategy in such issues [73]. Subsequently, the overall mechanisms enable AFs, which are particularly persistent, to remain active for long periods of time in the food, and on consumption they pose risks to human health [5,8].

Fumonisins, which can have an acidic nature, are produced by field fungi such as *Fusarium* (*F. moniliforme*, *F. verticillioides*, *F. proliferatum*.) These metabolites exhibit heat stability within the range of 100–120 °C and thrive under neutral pH conditions, allowing them to withstand common food industrial operations within this temperature range. These fungi and FBs largely infect cereal commodities, as stated in the previous section regarding their occurrence in 51.19% of samples. Particularly, FB contamination in individual products was 77.35% to 81.55% in cereals and nuts, together with corn products (Figure 1). The chemical structure of FBs includes aminopentol that has one tricarballylic acid on each side chain, along with OH groups [74,75]. Structurally, FBs also resemble sphinganine, and it is suggested that FBs may act as inhibitors of sphingolipid biosynthesis. Sphingolipids are essential chemicals of cell membranes and play a pivotal role in manifold signaling pathways [75]. The stability and lipophilic characteristics of FBs contribute to their persistence and accumulation in cereal matrices. Maximum accumulation of FBs has been associated with sphingolipids and oxylipin metabolism in corn grains, where these components are involved in fungus growth and FB production [76]. Moreover, the lipophilic nature of FBs allows them to react and accumulate within the lipid tissues. Consequently, cereal grains with higher lipid content are more susceptible to contamination. Like corn, other cereal grains, along with nuts rich in lipids, may be more prone to FB incidence. A variety of lipids in these foodstuffs can facilitate toxin production and fungal virulence [77]. The persistence of mycotoxins in cereal grains may also be attributed to mechanisms that fungi have evolved to protect themselves from their own toxins or to resist stress conditions, such as efflux pumps and transporters, respectively [78,79]. Similarly, ZEN as a non-polar, non-steroidal, and stable (up to ~120 °C) estrogen metabolite of field *Fusarium* (*F. graminearum*) that largely occurs in food and feed. The contamination of ZEN appeared in 40.25% of the overall evaluated products (Table 1). Relevant to its occurrence in individual food or feeds, ZEN is notable in animal-derived products and was found in in 99% of milk and 52.17% of chicken meat, while 51.48% of corn (Figure 1). These occurrences may be correlated to multiple factors, including those previously described. Moreover, ZEN and its conjugated ZEN-14-sulfate can commonly occur in cereal grains, and the bran and starchy fibrous portions of wheat grains were shown to have considerably high levels of these metabolites [66].

### 3.3. Genetic and Enzyme Involvement in Toxin Persistence in Food

Besides the stated mechanisms, it can be suggested that both genetic and enzymatic factors may also support toxin persistence and attachment to food components, leading to changes in gene expression that could potentially affect grain and quality. Research efforts have identified genes and enzymes which either positively or negatively regulate fungi-based mycotoxin biosynthesis in grains [80]. This involves up- or down-regulation of genes associated with mycotoxin contamination and persistence in foods. Table 2 displays only a few examples of such genes according to their host and mode of action, while several others have been studied so far. By manipulating these and other relevant genes, mycotoxin contamination and persistence in food may be minimized (Table 2). In wheat and corn, several genes have been found to be the main triggers in several mechanisms such as biosynthesis, stress response, and defense mechanisms against mycotoxins. The genetic basis of fumonisin resistance in corn kernels is complex, influenced by both genotype–environment interaction and multiple genes with small effects. Single-nucleotide polymorphisms (SNPs) have been shown to be associated with resistance and candidate genes involved in corn immune response signaling that represent opportunities for targeted functional variation to reduce kernel contamination without impacting beneficial behavior [81].

Mycotoxin persistence in cereals and nuts spans an array of factors, and the interaction occurs at several interfaces. It basically begins at the pre-harvest stage where the crops face fungal threats, which last, in the form of mycotoxins, for the rest of grains’ life to post-harvest and consumption stages. Although the underlying mechanism of interaction of the fungus or its mycotoxins with the host at the cellular and molecular level has been investigated, it is still unclear what encourages this persistence, especially at the interface between toxins and host components. Are there any active biomolecules present at the host product’s surface which promote mycotoxin persistence? If yes, what kind of interaction is there between those molecules and mycotoxins? Could these interactions be diminished or minimized and how can this be achieved? Studies addressing these gaps would aid in deciphering the persistence of mycotoxins at the molecular interface that would eventually lead to better remedies for maximum removal of mycotoxins, promoting safer food and contributing to the global “One Health” perspective.

## 4. Concluding Remarks

This review shed light on a major issue of fungi and mycotoxins in food/feeds. Additionally, some key details were presented on the metabolic activities that per our knowledge may be involved in a mycotoxin’s persistence in food grains in storage. Consistent with this study, mycotoxin control in food requires some advanced approaches, including the use of good agricultural practices—e.g., crop rotation, irrigation, fertilization, proper harvesting, handling and storage conditions, continuous monitoring, and post-harvest treatments like drying, cleaning, and sorting of the grains. However, these approaches may not be enough to prevent mycotoxins in cereal grains and even nuts, but may be possible by exploring the persistence issue in these items. Hence, state-of-the-art approaches like high-throughput “omics”, high-resolution imaging, and computational modeling [9,10,11] can be helpful. Besides the structural and physicochemical features, these operations can capably expose how metabolic activities occur in the stored grain’s embryo, endosperm, etc. and their interplay with storage fungi and mycotoxins on the cellular levels. Strategies such as developing crops that are genetically resistant to fungal infections, introducing genes that produce antifungal compounds, or selecting naturally resistant crop varieties can also reduce mycotoxin contamination and persistence, but the risks and benefits of these operations must be carefully evaluated to ensure they are safe for human health and the overall environment.

The scientific efforts proposed here can have significant implications, particularly in the development of high-throughput technologies for in-depth understanding of the potential issues related to storage fungi, mycotoxins, and their persistence in food on the cellular level. Profound details of the presented paradigm could allow further improvement in the operative procedures to mitigate mycotoxins in the food system on a large scale in the near future. This concept has the potential to greatly benefit decision-makers, food industrialists, professionals, and food scientists. In practice, this will require collaborative efforts among professionals from various disciplines, including governmental law enforcement bodies, academic researchers in food science and engineering, and experts in genetics, molecular and computational biology, as well as omics. This collaborative network and funding will enable the development of effective solutions for addressing the issues related to toxin persistence from genetic to molecular and metabolic levels in the food system.

## Figures and Tables

**Figure 1 foods-12-04314-f001:**
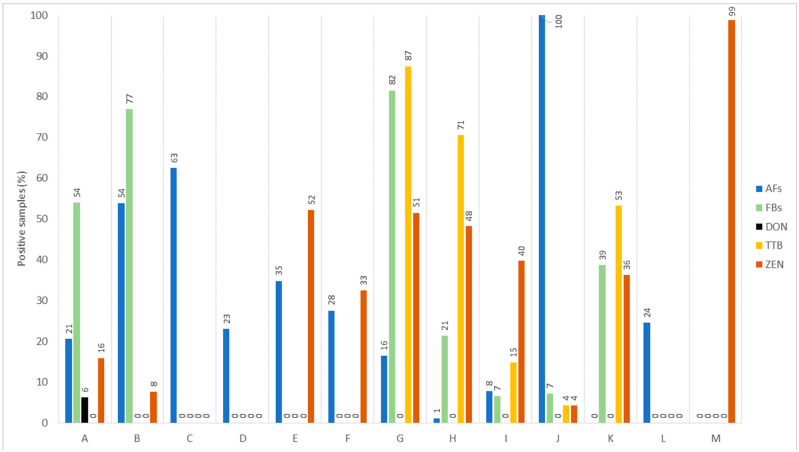
Overall percentage of positive samples of foods for mycotoxins recently reported (2017–2022). A, cereal-based infant foods; B, mix cereals and nut-based food; C, milk formula; D, fluid milk; E, chicken meat; F, eggs; G, corn as feed; H, wheat; I, soybean meal; J, peanut meal; K, oat grass; L, corn; M, dairy milk. AFs, aflatoxins; FBs, fumonisins; DON, deoxynivalenol; TTB, trichothecene type B; ZEN, zearalenone.

**Figure 2 foods-12-04314-f002:**
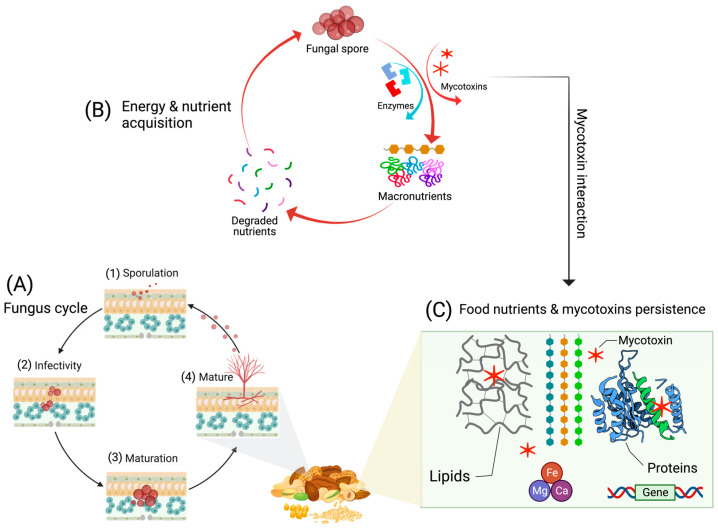
General illustration of the fungal life cycle (**A**) in food, starting with sporulation (1), infection (2), maturation (3) and ends with a mature fungus (4) on the food tissue. The amplified region (**B**) shows the secretion of the fungus hydrolytic enzymes that degrade macronutrients in the food and produce mycotoxins (highlighted with red stars) as defense chemicals in the intracellular space of the food tissue. Additionally, mycotoxins interact with food nutrients (**C**), and several genes may be involved that aid toxin interaction and continuous persistence.

**Table 1 foods-12-04314-t001:** Mycotoxin occurrence and levels in diverse foodstuffs recently reported in some countries.

Country/ Sampling Year	Type of Sample (*N*)	Mycotoxin	Positive Samples *n* (%)	Mean (Min–Max) µg/kg	Ref.
Nigeria					
2018–2021	Cereal-based infant food (63)	3-NPA	5 (7.9)	4.43 (1.24–7.01)	[30]
		AFB_1_	8 (12.7)	2.71 (0.36–14.4)	
		AFB_2_	3 (4.8)	0.42 (0.11–0.98)	
		AFG_1_	1 (1.6)	0.62 (0.62–0.62)	
		AFs	8 (12.7)	2.95 (0.36–15.4)	
		AFM_1_	1 (1.6)	0.57 (<LOD–0.57)	
		ALS	16 (25.4)	2.85 (1.12–5.97)	
		AME	11 (17.5)	1.49 (0.26–6.58)	
		ALT	6 (9.5)	4.71 (1.8–17.1)	
		BEA	44 (69.8)	0.29 (0.04–1.64)	
		Cereulide	5 (7.9)	1.07 (0.3–2.34)	
		CIT	14 (22.2)	106 (5.16–787)	
		DON	4 (6.3)	39.4 (3.14–110)	
		DHC	7 (11.1)	18.1 (1.88–54.5)	
		EnnA	3 (4.8)	0.24 (0.09–0.41)	
		EnnA_1_	7 (11.1)	0.47 (0.1–1.62)	
		EnnB	23 (36.5)	0.91 (0.02–8.14)	
		EnnB_1_	17 (27)	0.5 (0.06–2.61)	
		FB1	15 (23.8)	35.1 (3.99–66.2)	
		FB2	13 (20.6)	15.9 (3.54–34.9)	
		FB3	6 (9.5)	9.62 (<LOD–9.62)	
		FBs	15 (23.8)	52.7 (3.99–98.1)	
		MON	6 (9.5)	3.14 (2.53–6.19)	
		OTA	2 (3.2)	0.76 (<LOD–0.76)	
		ST	8 (12.7)	0.29 (0.13–0.96)	
		TEN	4 (6.3)	1.63 (0.96–2.99)	
		ZEN	10 (15.9)	1 (0.32–3.22)	
	Mixed cereal- and nut-based foods (13)	3-NPA	5 (38.5)	10.1 (4.28–28.2)	
		AFB_1_	7 (53.8)	5.27 (0.36–14.3)	
		AFB_2_	4 (30.8)	1.84 (0.33–3.02)	
		AFG_1_	3 (23.1)	0.48 (0.27–0.89)	
		AFs	7 (53.8)	6.53 (0.36–17.4)	
		AFM_1_	3 (23.1)	0.56 (0.51–0.66)	
		ALS	7 (53.8)	1.62 (1.19–3.42)	
		AME	1 (7.7)	0.26 (<LOD–0.26)	
		ALT	1 (7.7)	1.8 (<LOD–1.8)	
		BEA	13 (100)	0.3 (0.04–0.76)	
		CIT	9 (69.2)	20.1 (6.81–50.2)	
		DON	0	0	
		DHC	4 (30.8)	20.3 (1.88–38.2)	
		EnnA	0	0	
		EnnA_1_	2 (15.4)	0.26 (0.1–0.43)	
		EnnB	5 (38.5)	0.46 (0.06–1.13)	
		EnnB_1_	3 (23.1)	0.3 (0.06–0.51)	
		FB_1_	9 (69.2)	27.2 (14.3–60.3)	
		FB_2_	9 (69.2)	6.7 (3.54–15.3)	
		FB_3_	1 (7.7)	9.62 (<LOD–9.62)	
		FBs	10 (76.9)	31.5 (10.6–85.2)	
		MON	8 (61.5)	6.72 (2.53–28.6)	
		OTA	1 (7.7)	2.65 (<LOD–2.65)	
		ST	2 (15.4)	0.85 (0.73–0.97)	
		TEN	1 (7.7)	0.4 (<LOD–0.4)	
		ZEN	1 (7.7)	0.8 (0.8–0.8)	
Chile					
2018–2020	Cocoa (22)	AFs	0	ND	[31]
		OTA	7 (31.8)	N.I. (ND–4.77)	
	Oat (50)	AFs	0	ND	
		OTA	1 (2)	N.I. (ND–1.74)	
	Cereals (60)	AFs	0	ND	
		OTA	1 (1.6)	N.I. (ND–2.51)	
	Peanuts (35)	AFs	0	ND	
		OTA	0	ND	
	Hazelnut (7)	AFs	0	ND	
		OTA	0	ND	
2022	Milk formula (24)	AFM_1_	15 (62.5)	0.0038 (0.006–0.0117)	[32]
	Fluid milk (26)	AFM_1_	6 (23.1)	0.0069 (0.0063–0.0075)	
	Chicken meat (115)	AFs	40 (34.8)	2.4 (<LOD–8.01)	[18]
		OTA	47 (40.9)	1.14 (<LOD–4.7)	
		ZEN	60 (52,2)	2.01 (<LOD–5.01)	
	Eggs (80)	AFs	22 (27.5)	1.97 (<LOD–4.46)	
		OTA	28 (35)	1.17 (<LOD–2.98)	
		ZEN	26 (32.5)	1.58 (<LOD–3.6)	
China					
2017–2021	Corn as feed (2873)	AFs	474 (16.5)	63.28 (N.I.–773)	[17]
		TTB	2513 (87.47)	871.28 (N.I.–12,808)	
		FBs	2343 (81.55)	2618.81 (N.I.–40,090)	
		ZEN	1479 (51.51)	176.79 (N.I.–4686)	
	Wheat (411)	AFs	5 (1.22)	2.6 (N.I.–5)	
		TTB	290 (70.56)	2129.29 (N.I.–59,325)	
		FBs	88 (21.41)	332.31 (N.I.–910)	
		ZEN	192 (48.18)	105.5 (N.I.–1205)	
	Soybean meal (257)	AFs	20 (7.78)	4.65 (N.I.–35)	
		TTB	38 (14.79)	171.5 (N.I.–597)	
		FBs	17 (6.61)	760.24 (N.I.–6932)	
		ZEN	102 (39.69)	45.26 (N.I.–237)	
	Peanut meal (69)	AFs	69 (100)	417.72 (N.I.–10,091)	
		TTB	3 (4.35)	77.67 (N.I.–139)	
		FBs	5 (7.25)	50.4 (N.I.–120)	
		ZEN	3 (4.35)	37.33 (N.I.–61)	
	Oat grass (124)	AFs	0 (0)	-	
		TTB	66 (53.23)	1,728.85 (N.I.–9363)	
		FBs	48 (38.68)	381.76 (N.I.–1986)	
		ZEN	45 (36.29)	484.53 (N.I.–2622)	
Serbia					
2018	Corn (100)	AFs	8 (8)	3.6 (0.8–8.3)	[33]
2019	Corn (100)	AFs	11 (11)	3 (0.6–10.9)	
2020	Corn (100)	AFs	5 (5)	2.1 (1.1–3.0)	
2021	Corn (100)	AFs	84 (84)	38.8 (0.5–246.3)	
Croatia					
2018	Corn (110)	AFs	15 (13.6)	6.2 (1.6–75.1)	
2019	Corn (109)	AFs	17 (15.6)	2.5 (1.5–26.9)	
2020	Corn (103)	AFs	20 (19.4)	1.6 (1.5–3.3)	
2021	Corn (111)	AFs	44 (39.6)	34.1 (1.5–422.2)	
Island of São Miguel (Portugal)					
2020	Dairy milk (22)	ZEN	22 (100)	3.53 (1.23–>4.46)	[34]
	Dairy milk (10)	ZEN	10 (100)	1.15 (0.48–2.15)	
	Dairy milk (27)	ZEN	26 (96.3)	0.48 (<LOD–1.37)	
	Dairy milk (25)	ZEN	25 (100)	1.15 (0.31–2.43)	

ND, not detected; N.I., not informed; LOD, limit of detection; 3-NPA, 3-nitropropionic acid; AFB_1_, aflatoxin B_1_; AFG_1_, aflatoxin G_1_; AFs, aflatoxins; AFM_1_, aflatoxin M_1_; ALS, altenuisol; AME, alternariol monomethyl ether; ALT, altersetin; CIT, citrinin; DON, deoxynivalenol; DHC, dihydrocitrinone; EnnA, enniatin A; EnnA_1_, enniatin A_1_; EnnB, enniatin B; EnnB_1_, enniatin B_1_; FBs, fumonisins; FB_1_, fumonisin B_1_; FB_2_, fumonisin B_2_; FB_3_, fumonisin B_3_; MON, moniliformin; OTA, ochratoxin A; ST, sterigmastocystin; TEN, tentoxin; TTB, trichothecene type B; ZEN, zearalenone.

**Table 2 foods-12-04314-t002:** An overview of some important genes involved in fungal infectivity of cereals and nuts.

Target Genes	Host	Mode of Action	Ref.
*PRms*	Corn	Positively regulates resistance against *A. flavus* infection and aflatoxin contamination	[82]
*WRKY*	Peanut	Mitigates aflatoxin production	[83]
*TaBCC3*	Wheat	Positive regulator of DON	[84]
*UDP-Glucosyl-transferase*	Wheat	Positively regulates resistance against Fusarium head blight (FHB)	[85]
*P450 lanosterol 14α-demethylase (CYP51)*		Works against DON contamination in host-induced gene silencing (HIGS) lines	
*TaCYP72A*	Wheat	Positive regulator of DON	[86]
*TaNFX1*	Wheat	Negative regulator of resistance against Fusarium head blight	[87]

## Data Availability

The data used to support the findings of this study can be made available by the corresponding author upon request.

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
