# Peer review of "Occurrence of Mycotoxins in Foods: Unraveling the Knowledge Gaps on Their Persistence in Food Production Systems"

_foods, 2023, doi:10.3390/foods12234314_

Round 1

Reviewer 1 Report

Comments and Suggestions for Authors

The objective of this Review (ID:foods-2737887) entitled “Occurrence of Mycotoxins in Foods: Unraveling the Knowledge Gaps on Their Persistence in Food Production Systems  was to explore the intricate issue about the occurrence levels of mycotoxins in foods, to underline the main knowledge gaps on the persistence of these contaminates in the food production system.

The topic of this review is very important regarding food safety, considering the increasing frequency of extreme weather events and the global climate changes that affect the frequent occurrence of mycotoxins in food and feed worldwide.

In my opinion, this review paper needs to be supplemented and improved.

There are a lot of points that have to be clarified and corrected, which I specified below:

1.      Authors should supplement and improve the section “2. Mycotoxin Occurrence in Foodstuffs”, because numerous results have been published in the past five years regarding the occurrence of mycotoxins in food.

2.      In the legend of Table 1, the abbreviation PS should be deleted because it is not used in the manuscript

3.      Page 5, Lines 106-107, the sentence “The incidence and contamination rates (percentage of positive samples) can vary according to the sample size.” should be deleted.

4.       Fig. 1, in Legend stands for “Overall percentage of positive samples of foods for mycotoxins recently reported (2017-2023)”, while on Page 5, Lines 106-109 stands for “The incidence and contamination rates (percentage of positive samples) can vary according to the sample size. Regarding the overall samples evaluated in studies conducted between 2017 and 2022 and detailed in Table 1, higher contamination rates were observed for trichothecene type B (TTB) toxins (58.53%), FBs (51.19%), ZEN (40.25%) and AFs (17%), as depicted in Figure 1.”

Does the data in Table 1 and Figure 1. refer to the period 2017-2022 or the period 2017-2023?

5.      Page 7, Lines 184-185 the sentence sounds weird. The authors should reformulate it.

6.      Authors should use either “corn” or “maize” throughout the manuscript.

7.      Aspergillus spp. are usually considered to be storage fungi, but aflatoxins may be synthesized under field conditions. This is evidenced by numerous publications and the data shown in Table 1 and Figure 1. Consequently, on  Page 9, Lines 251-255, the authors should also add Aspergillus spp.

Author Response

Reviewer #1

The objective of this Review (ID:foods-2737887) entitled “Occurrence of Mycotoxins in Foods: Unraveling the Knowledge Gaps on Their Persistence in Food Production Systems”  was to explore the intricate issue about the occurrence levels of mycotoxins in foods, to underline the main knowledge gaps on the persistence of these contaminates in the food production system. The topic of this review is very important regarding food safety, considering the increasing frequency of extreme weather events and the global climate changes that affect the frequent occurrence of mycotoxins in food and feed worldwide. In my opinion, this review paper needs to be supplemented and improved. Response: The authors greatly appreciate the reviewer, and they hope to have properly addressed all possible comments in the revised version of the manuscript.

There are a lot of points that have to be clarified and corrected, which I specified below:

  1. Authors should supplement and improve the section “2. Mycotoxin Occurrence in Foodstuffs”, because numerous results have been published in the past five years regarding the occurrence of mycotoxins in food.

Response: Upon consideration, the authors would like to emphasize that the focus of this section is to provide an overview on the “OCCURRENCE” of mycotoxins in foodstuffs. The intention is to utilize this information to showcase as a supportive element for the central concept of "PERSISTENCE" explored in the manuscript. While acknowledging that there have been numerous research findings on mycotoxin occurrence in the past years, and expanding this section further might deviate from the manuscript's original emphasis on “PERSISTENCE”. Adding more details to the OCCURRENCE aspect could potentially dilute the clarity and coherence of the core argument of “PERSISTENCE”.

  1. In the legend of Table 1, the abbreviation PS should be deleted because it is not used in the manuscript

Response: Thanks, PS was deleted from the table’s caption.

  1. Page 5, Lines 106-107, the sentence “The incidence and contamination rates (percentage of positive samples) can vary according to the sample size.” should be deleted.

Response: The recommended line was deleted.

  1. Fig. 1, in Legend stands for “Overall percentage of positive samples of foods for mycotoxins recently reported (2017-2023)”, while on Page 5, Lines 106-109 stands for “The incidence and contamination rates (percentage of positive samples) can vary according to the sample size. Regarding the overall samples evaluated in studies conducted between 2017 and 2022 and detailed in Table 1, higher contamination rates were observed for trichothecene type B (TTB) toxins (58.53%), FBs (51.19%), ZEN (40.25%) and AFs (17%), as depicted in Figure 1.”

Does the data in Table 1 and Figure 1. refer to the period 2017-2022 or the period 2017-2023?

Response: Thanks for the comment, the data year “2023” was corrected as “2022”.

  1. Page 7, Lines 184-185 the sentence sounds weird. The authors should reformulate it.

Response: The recommended lines were rewritten.

  1. Authors should use either “corn” or “maize” throughout the manuscript.

Response: These interchanged terms were replaced to “corn”.

  1. Aspergillus spp. are usually considered to be storage fungi, but aflatoxins may be synthesized under field conditions. This is evidenced by numerous publications and the data shown in Table 1 and Figure 1. Consequently, on Page 9, Lines 251-255, the authors should also add Aspergillus spp.

Response: The authors thank the reviewer; however, the mentioned lines specifically focus on field fungi, and the inclusion of Aspergillus spp. as storage fungi may not align with the intended scope of that section (lines 251-255). Those details on Aspergillus spp., particularly in their role as storage fungi, are indeed covered in the subsequent text on pages 10-11.

Reviewer 2 Report

Comments and Suggestions for Authors

The manuscript Occurrence of Mycotoxins in Foods: Unraveling the Knowledge Gaps on Their Persistence in Food Production Systems, submitted for review, is, in the reviewer's opinion, very interesting. The authors have attempted to clarify this complex but very important topic for public health.

The manuscript shows signs of originality and is an overview of the global situation. I have no reservations about the construction and description of the article.

In the course of reading the article, a considerable amount of self-quotation of Carlos Augusto Fernandes de Oliveira (7x) was noted, please reduce this.
Please revise References with requirements, whether the list is to be arranged alphabetically or by order of citation in the text.

Author Response

Reviewer #2

The manuscript Occurrence of Mycotoxins in Foods: Unraveling the Knowledge Gaps on Their Persistence in Food Production Systems, submitted for review, is, in the reviewer's opinion, very interesting. The authors have attempted to clarify this complex but very important topic for public health. The manuscript shows signs of originality and is an overview of the global situation. I have no reservations about the construction and description of the article.

Response: Thanks very much for your constructive comments.

In the course of reading the article, a considerable amount of self-quotation of Carlos Augusto Fernandes de Oliveira (7x) was noted, please reduce this. Please revise References with requirements, whether the list is to be arranged alphabetically or by order of citation in the text.

Response: Thank you for your feedback. The authors acknowledge the concern about self-citation, and after careful evaluation, the self-citation percentage was confirmed “7.7%”, well below the journal's recommended threshold of 15%. In response to the reference (text & bibliography) organization, all Refs. were reordered (as highlighted in yellow). The authors believe these revisions address your concerns, and they remain open to any additional suggestions or modifications you may propose.

Round 2

Reviewer 1 Report

Comments and Suggestions for Authors

I appreciate the authors for their responses to my comments. 

The manuscript can be accepted in its current form.